# Brown Adipose Tissue Sheds Extracellular Vesicles That Carry Potential Biomarkers of Metabolic and Thermogenesis Activity Which Are Affected by High Fat Diet Intervention

**DOI:** 10.3390/ijms231810826

**Published:** 2022-09-16

**Authors:** Tamara Camino, Nerea Lago-Baameiro, Aurelio Sueiro, Susana Belén Bravo, Iván Couto, Francisco Fernando Santos, Javier Baltar, Felipe F. Casanueva, María Pardo

**Affiliations:** 1Grupo Obesidómica, Área de Endocrinología, Instituto de Investigación Sanitaria de Santiago de Compostela (IDIS), Xerencia de Xestión Integrada de Santiago (XXIS/SERGAS), 15706 Santiago de Compostela, Spain; 2Grupo Endocrinología Molecular y Celular, Instituto de Investigación Sanitaria de Santiago (IDIS), Xerencia de Xestión Integrada de Santiago (XXIS/SERGAS), 15706 Santiago de Compostela, Spain; 3Unidad de Proteómica, Instituto de Investigación Sanitaria de Santiago (IDIS), Xerencia de Xestión Integrada de Santiago (XXIS/SERGAS), 15706 Santiago de Compostela, Spain; 4Servicio de Cirugía Plástica y Reparadora, Xerencia de Xestión Integrada de Santiago (XXIS/SERGAS), 15706 Santiago de Compostela, Spain; 5Servicio de Cirugía General, Xerencia de Xestión Integrada de Santiago (XXIS/SERGAS), 15706 Santiago de Compostela, Spain; 6CIBER Fisiopatología Obesidad y Nutrición, Instituto de Salud Carlos III, 28029 Madrid, Spain

**Keywords:** obesity, extracellular vesicles, exosomes, adipose tissue, white adipose tissue, brown adipose tissue, batosomes, proteome, metabolic diseases, cell communication

## Abstract

Brown adipose tissue (BAT) is a key target for the development of new therapies against obesity due to its role in promoting energy expenditure; BAT secretory capacity is emerging as an important contributor to systemic effects, in which BAT extracellular vesicles (EVs) (i.e., batosomes) might be protagonists. EVs have emerged as a relevant cellular communication system and carriers of disease biomarkers. Therefore, characterization of the protein cargo of batosomes might reveal their potential as biomarkers of the metabolic activity of BAT. In this study, we are the first to isolate batosomes from lean and obese Sprague–Dawley rats, and to establish reference proteome maps. An LC-SWATH/MS analysis was also performed for comparisons with EVs secreted by white adipose tissue (subcutaneous and visceral WAT), and it showed that 60% of proteins were exclusive to BAT EVs. Precisely, batosomes of lean animals contain proteins associated with mitochondria, lipid metabolism, the electron transport chain, and the beta-oxidation pathway, and their protein cargo profile is dramatically affected by high fat diet (HFD) intervention. Thus, in obesity, batosomes are enriched with proteins involved in signal transduction, cell communication, the immune response, inflammation, thermogenesis, and potential obesity biomarkers including UCP1, Glut1, MIF, and ceruloplasmin. In conclusion, the protein cargo of BAT EVs is affected by the metabolic status and contains potential biomarkers of thermogenesis activity.

## 1. Introduction

Obesity has been established as a predominant disease worldwide. Excess body weight and chronic exposure to an obese state are associated with inflammation, which is is exacerbating and/or promoting a plethora of pathologies that are putting health systems under pressure; moreover, the COVID-19 pandemic has brought to the forefront the higher risk of a fatal outcome for those people with excess body weight and associated pathologies such as type 2 diabetes [1].

Under this context, there is no doubt that there is a necessity to improve our knowledge about the mechanisms implicated in the development of obesity and its comorbidities, with the objective to tackle further harm when intervention with educational preventions is not successful. Although there has certainly been impressive progress in this regard, there are still many unexplored aspects regarding the development of obesity. It is well known that different organs and tissues that are involved in metabolism at the peripheral and central levels suffer important deregulations at the onset of obesity including altered secretion of hormones and cytokines. Among these deregulations, deregulated secretion of adipose tissue as a consequence of hypertrophy, hyperplasia, ECM stiffness, and inflammation linked to energy intake excess have still not been clearly elucidated; thus, new key players are coming into light such as the secretion of sophisticated messengers assembled in extracellular vesicles [2].

Relatively recently, extracellular vesicles (EVs) have emerged as new important players in physiological and pathological situations with respect to communication among cells and tissues at the paracrine and endocrine levels. These vesicles are small round-shaped membrane spheres of various sizes (30–1000 nm) shed by virtually all cell types; therefore, they can be found in the extracellular surroundings and in all body fluids including blood, urine, cerebrospinal fluid, saliva, tears, etc. According to their biogenesis and size, they can be classified as microvesicles (100 nm–1 um) derived by blebbing of the plasma membrane, exosomes (30–100 nm) compiled in multivesicular endosomes (MVEs) that are secreted by exocytosis, and larger vesicles (50–5000 nm) that comprise apoptotic bodies liberated by cells prior to apoptosis [3,4]. The interesting aspect of EVs is that they carry membrane and cytosolic components such as proteins, lipids, and RNAs, and this molecular cargo is dynamically affected by the site of biogenesis, the metabolic status, as well as the physiological/pathological conditions. Therefore, EVs have also emerged as key potential biomarkers of pathology, which can provide significant information through low invasive approaches.

Under this context, recently, it has been described that adipose tissue EVs play a role in intercellular and inter-organ crosstalk in metabolic health and diseases. Precisely, EVs that are shed by adipose tissue cell components such as adipocytes, mesenchymal stem cells, endothelial cells, and immune cells actively participate in the modulation of adipogenesis, browning of adipose tissue, adipokine release, the immune microenvironment, and tissue remodeling (reviewed in [5,6]). Moreover, EVs from adipose tissue are able to exert metabolic actions and spread pathology locally [7], and also at a distance level, by interacting with organs implicated in the regulation of energy metabolism and insulin sensitivity such as skeletal muscle, liver, pancreas, and brain [8]. Thus, upon interaction with target cells, EVs exert signal transduction or regulate gene expression by post-transcriptional regulation throughout miRNAs, mRNAs, and lncRNAs [6]. Under this context, there is robust evidence that has demonstrated the presence of EVs in the culture medium of human and murine adipose tissue explants [9,10] and also in the cell culture medium of primary cultured adipocytes [11] and preadipocyte-differentiated murine cells [12,13]. Importantly, there is increasing evidence that has implicated EVs in obesity-associated metabolic deregulation including its comorbidities [14,15], and more precisely, in the local and systemic inflammation linked to liver and adipose tissue [8]. Thus, EVs shed by adipose tissue have been proposed to be involved in adipocyte/macrophage crosstalk [16,17] and to affect insulin signaling and expression in muscle and liver cells leading to metabolic disease [18,19,20]. Moreover, it has been shown that EVs secreted by tumor resident adipocytes interacted with melanoma cells and induced migration and invasion through fatty acid oxidation, especially in obese animals [12].

There are only a few studies that have focused on EVs secreted by white adipose tissue, and only anecdotal knowledge is available about those shed by brown adipose tissue (BAT). BAT is the major site of adaptive thermogenesis, and several studies have associated BAT activity with protection against obesity and metabolic diseases, such as type 2 diabetes mellitus and dyslipidemia. Active BAT is present in adult humans and its activity is compromised in patients with obesity. The capacity of BAT to protect against chronic metabolic disease has traditionally been attributed to its ability to use glucose and lipids for thermogenesis. Yet, it has been suggested that BAT might also have a secretory role, which could contribute to the systemic effects of BAT activity [21]. Thus, in vitro studies have demonstrated that the amount of EVs secreted by beige and brown adipocytes can be increased by treatment with cAMP, the second messenger induced by cold exposure or beta-adrenergic stimulation [22].

Taking into consideration all the abovementioned points, there is a need to discern the regulation of EVs’ cargo composition, its alteration caused by obesity, the mechanism of tropism, and to get deeper knowledge about the functional role of EVs upon interaction with target cells/tissues [6]. Thus, in this research, we performed an extensive analysis of those vesicles secreted by adipose tissue at different anatomical locations elucidating the EVs cargo changes after high fat diet (HFD) intervention for 12 weeks. Thus, we isolated extracellular vesicles shed by brown adipose tissue in the same animals, and described, for the first time, the reference proteome map of batosomes and, additionally, performed a quantitative proteomic analysis using SWATH for comparisons with white adipose tissue-shed vesicles (subcutaneous and visceral depots) from the same animals under lean and obese conditions. A functional analysis and proteins of interest validation reveals, for the first time, the protein cargo dynamics of batosomes during the development of obesity.

## 2. Results

### 2.1. White and Brown Adipose Tissues Shed EVs Which Change Their Tetraspanins Profile after High Fat Diet (HFD) Intervention

We established an HFD (60% fat, 9 weeks) animal model of obesity to analyze the effect of obesity on the protein content of white and, especially, brown adipose tissue shed extracellular vesicles (EVs). Once established, BMI (body mass index) and fasting glucose showed significant variations among lean and obese animals in this study (Figure 1A,B); thus, explants of visceral (VAT), subcutaneous (SAT) and brown (BAT) adipose tissues were extracted and cultured to isolate specific EVs through serial ultracentrifugation or by using a Nanoview platform (ExoView analysis), following our own established protocols [7,9] (Figure 1C). Characterization of isolated vesicles through serial centrifugation showed no significant differences in vesicle size depending on the depot of origin nor comparing lean and obese animals, ranging from 75 to 112 nm by applying a nanoparticle tracking analysis (NTA) (Figure 1D). 

Complete EV characterization using an ExoView platform allowed us to perform particle sizing, EV count, and a single vesicle phenotype analysis of secreted vesicles from each adipose tissue using functionalized microchips (Figure 2). Thus, the secretome samples were directly incubated in individual chips containing anti-CD81 and -CD9 as capture antibodies; further, once the EVs were immunocaptured, anti-CD63, -CD9, and -CD81 labeled antibodies were used for detection to perform a tetraspanins profile of each EV depot (Figure 2A). In general, we observed that EVs captured by CD9 and CD81 did not present statistically significant differences in size, which measured between 55 and 75 nm (Figure 2B,C). However, it was observed that the numbers of particles captured by CD9 and CD81 were statistically significantly higher in EVs from VAT and BAT of normal weight rats than in vesicles from the same depots in obese animals, and also as compared with those vesicles of subcutaneous origin from both lean and obese animals (Figure 2D). Furthermore, globally, it was found that EVs captured by CD9 and CD81 showed more CD63 positive vesicles in lean SAT than lean VAT; moreover, an augmentation of CD63 positive vesicles were found on all the obese tissues (SAT, VAT and BAT), especially in obese BAT vesicles (Figure 2E). There was also a diminution of CD81 positive vesicles with obesity, especially in obese BAT. In addition, lean SAT and VAT vesicles contained fewer CD9 positive vesicles than BAT, and an increment of CD9 positive vesicles was observed with obesity in VAT and BAT (Figure 2E).

Thus, it can be concluded that white and brown adipose tissues shed EVs with similar size independently of the depot or the metabolic status; however, they change their tetraspanins profile after HFD intervention.

### 2.2. Proteome Content of White and Brown Adipose Tissue Shed EVs Shows Depot-Specific Protein Profiles That Reflect Metabolic Deregulation in Those Animals with HFD Intervention

A mass spectrometry qualitative DDA proteomic analysis of EVs isolated from four independent obese animals and four independent lean animals was performed that showed different protein profiles for SAT, VAT, and BAT vesicles depending on the location of fat, which were influenced by HFD (Figure 3A,B and Appendix A). The number of identified proteins for each EV depot in lean and obese states are shown in Figure 3B. The functional enrichment analysis of all proteins showed that most of the identified proteins were present in the Vesiclepedia database, as expected (Appendix A). Among all identified proteins, 98 proteins were common to all lean vesicles independently of the fat depot, 108 proteins were common to all obese vesicles, and 70 proteins were common to all EVs (Appendix A). A functional analysis of the proteins common to all analyzed vesicles showed that a high percentage of proteins were classified as related to metabolism, energy pathways, cell growth and maintenance, nucleotide and protein metabolism, and transport (Appendix A). Interestingly, we found functional classification differences in those proteins exclusively identified in adipose tissue-secreted vesicles of lean animals as compared with obese animals; the lean animals’ EVs contained more proteins related to metabolism and energy pathways, protein metabolism, and cell growth, whereas the obese animals’ vesicles carried more signal transduction-related proteins, cell communication, and more interestingly, those classified as immune response proteins. Moreover, when analyzing proteins in biological pathways, obese EVs stood out for syndecan-mediated signaling events, the PDGF receptor signaling network, beta1 integrin–cell surface interactions, Arf6 trafficking events, plasma membrane estrogen receptor signaling, PAR1-mediated signaling events, and coherently, by a higher percentage of proteins associated with the insulin pathway (Appendix A).

By comparing the identified proteins in the three depots of obese and lean animals independently, we observed that the exclusive proteins for each type of tissue (BAT, SAT, and VAT) in obese animals revealed that obese BAT EVs contained a higher percentage of proteins related to metabolism and energy pathways than those exclusively found in obese SAT and VAT vesicles (Figure 3C); obese SAT EVs were characterized by proteins involved in cell communication and signal transduction, and obese VAT EVs were characterized by proteins involved in cell growth and maintenance and protein metabolism (Figure 3C). Concerning biological pathway classification, again, obese BAT vesicles stand out with a high percentage of metabolism-associated proteins, as well as proteins related to fatty acid, TG, and ketone body metabolism, and to metabolism of lipids and lipoproteins (Figure 3D). Obese VAT vesicle exclusive proteins were found to be associated with signaling events mediated by VEGFR1/2, beta1 integrin–cell surface interactions, Class I PI3K signaling events, and integrin family–cell surface interactions (Figure 3D). Interestingly, when comparing the exclusive proteins identified in the three depots of lean animals, we found similar differences according to the biological process (Figure 3E). Something similar was found when classifying proteins according to biological pathway; however, some differences showed that lean BAT vesicles contained exclusive proteins related to the citric acid (TCA) cycle and respiratory electron transport as compared with lean SAT and VAT vesicles. In addition, lean white depot-secreted vesicles (SAT and VAT) showed proteins associated with alpha9beta1 integrin signaling events, LKB1 signaling events, the glypican pathway, and ErbB1 downstream signaling, at a much higher percentage as compared with lean BAT EVs (Figure 3F).

An analysis of the proteins found exclusively on the vesicles shed by the same depot of lean vs. obese animals showed that obese SAT vesicles carried a higher percentage of proteins related to cell communication, signal transduction, and cell growth than lean SAT; on the contrary, the lean SAT vesicles were characterized by protein metabolism, energy pathways, and metabolism-related proteins (Appendix A). In addition, the differences seemed even greater when studying protein in vesicles secreted by visceral adipose tissue. Exclusive proteins in EVs from obese VAT were classified as participating in protein metabolism and cell growth in relation to biological processes; moreover, they contained a significantly elevated percentage of proteins related to insulin, IGF1, TRAIL, syndecan, estrogen receptor, IL-5, among others signaling pathways (Appendix A).

By comparing the DDA-identified proteins with our previous work on human obese VAT- and SAT-secreted vesicles, we detected 131 proteins identified in obese rat VAT EVs that were common to those identified in obese human VAT EVs (Appendix A); and among those, FABP4, Histone H4, Thy-1 membrane glycoprotein, and vimentin were found to be elevated in human obese VAT vesicles in our previous analysis [9]. Moreover, 125 proteins were common to obese rat SAT EVs and human obese SAT vesicles (Appendix A); among those, 4F2 cell surface antigen heavy chain, fibrinogen beta and gamma chain, heat shock 71 kDa protein, lactadherin, lysosome-associated membrane glycoprotein (LAMP-1), and programmed cell death 6 interacting protein (ALIX) were also elevated in human obese SAT-secreted EVs in our previous report [9]. We additionally compared the protein cargo of rat obese SAT and VAT EVs with our results regarding in vitro murine adipocyte cell models of insulin resistance and lipid hypertrophy [7]. On the one hand, we found that obese rat VAT EVs shared nine proteins with those identified in insulin resistant vesicles, and three proteins with those from lipid hypertrophied adipocytes (Appendix A); on the other hand, obese rat SAT EVs shared 13 proteins with insulin resistant adipocyte-shed EVs, and eight proteins with lipid hypertrophied (Appendix A). It is worth highlighting that, from our previous postulated obese adipose EV biomarkers list [5,7,9], we could detect FABP4, CD36 antigen, and ceruloplasmin both in obese rat VAT vesicles and SAT vesicles (Appendix A).

While, in this study, we isolated BAT-shed EVs, which, to the best of our knowledge, have not been analyzed earlier, we paid special attention to the mass spectrometric results of these vesicles. First, we found that 60% of the cargo proteins in brown adipose tissue EVs were unique to this type of tissue as compared with those secreted by white depots (VAT and SAT); most of them were classified as originiating from exosomes and components of mitochondria (Figure 3A,B). Based on a comparison of the DDA-identified proteins with the BATLAS database based on transcriptome signatures of murine brown, brite(beige), and white adipocytes [23]; we identified 24 common proteins (Appendix A). Interestingly, we observed differences in the protein profiles of these particles depending on the metabolic state of the animal (Appendix A); thus, we found a common pattern in the BAT EVs secreted by obese animals, which was that a higher percentage of proteins were classified as related to protein metabolism and syndecan signaling events; moreover, there was an increase in the proteins related to integrin family–cell surface interactions and ErB1 signaling. Again, in a similar way to what was found in lean VAT and SAT vesicles, the vesicles secreted by lean BAT, were enriched in proteins related to metabolism and energy pathways (Appendix A).

To obtain a better picture of the protein cargo of EVs from lean and obese adipose tissues, we performed a differential quantitative analysis by DIA sequential window acquisition of all theoretical fragment-ion spectra (SWATH) label-free proteomics; hence, we aimed to discern the enriched or diminished proteins in EVs associated with obesity metabolism deregulation. A principal component analysis (PCA) of the differentially expressed proteins across tissues and metabolic status (lean and obese) revealed a clear separation of the vesicles according to the anatomical location of the fat depot; the separation of obese and lean EVs secreted by BAT was especially striking (Figure 3G).

The analysis of the differential expression of proteins with a fold change >1.5 allowed us to select the proteins that best differentiated the EVs from adipose tissue of lean rats as compared with EVs of the same tissue from obese animals. The list of all identified differences with significant *p*-values and fold changes higher than 1.5 is shown in Appendix A, and a selection of those with fold changes higher than 2 is shown in Table 1 and Figure 4, Figure 5 and Figure 6. Moreover, a list showing quantitative analysis of the differences among obese vesicles as compared with those secreted by lean animals in the three analyzed depots, is also described (Appendix A).

From the SWATH analysis we observed that in the case of EVs from subcutaneous adipose tissue, elevated proteins such as caveolin-1 (CAV1_RAT), vesicle-associated membrane protein 2 (VAMP2_RAT), and monoglyceride lipase (MGLL_RAT), among others, were observed in EVs from lean rats; in contrast to fibulin-5 (FBLN5_RAT), pyruvate dehydrogenase kinase isozyme 1 mitochondrial (PDK1_RAT) and pyruvate kinase PKM (KPYM_RAT) proteins in EVs from the same tissue of obese animals (Figure 4A,C,E and Table 1). In addition, functional analysis (FunRich database) of proteins elevated in SAT EVs showed the exosomal nature of these vesicles, especially those released from lean animals; 82% of the elevated proteins from lean SAT EVs belonged to the cytoplasm, in contrast to obese EVs in which 56% belonged to the plasma membrane. It is worth highlighting that the elevated proteins in obese SAT EVs were related to transport, lipid metabolism, and the immune response unlike the elevated proteins in EVs secreted from lean SAT which were related to lipid metabolism. In addition, 50% of the elevated proteins in lean SAT vesicles were related to metabolism as compared with 18% of those of obese subcutaneous origin (Figure 4F).

Additionally, proteins such as plastin-3 (PLST_RAT), destrin (DEST_RAT), or ATP synthase subunit beta (ATPB_RAT) were found to be elevated in lean rat VAT EVs; unlike carboxypeptidase Q (CBPQ_RAT), voltage-dependent anion-selective channel protein 2 (VDAC2), or protein S100-A4 (S10A4_RAT), among others, elevated in EVs from obese visceral adipose tissue (Figure 4B,D,E and Table 1). In addition, the functional analysis in the FunRinch database showed that EVs released from obese visceral fat contained a higher percentage of upregulated proteins that were associated with the plasma membrane, cytoplasm, and nuclei than EVs released from the same tissue of lean animals. In addition, the upregulated proteins in the VAT EVs of obese rats were related to the transport and TRAIL signaling pathway, unlike the lean EVs, that were related to metabolism (Figure 4F).

Moreover, it is important to highlight that, among the upregulated proteins in EVs released by brown adipose tissue from obese animals, there were mitochondrial brown fat uncoupling protein 1 (UCP1_RAT), macrophage migration inhibitory factor (MIF_RAT), ceruloplasmin (CERU_RAT), dipeptidase-1 (DPEP1), and CD14 (CD14_RAT) as compared with EVs from the same tissue of lean rats in which elevated proteins such as mitochondrial cytochrome b-c1 complex subunit 6 (QCR6_RAT) and ATP citrate synthetase (ACLY_RAT) were identified (Figure 5A,B and Figure 6 and Table 1). A comparative SWATH analysis of the proteins identified in BAT vesicles from obese animals as compared with all the other EVs (lean BAT, obese/lean SAT, and VAT) showed that UCP-1 protein was elevated in these vesicles six times with HFD intervention (Appendix A). In addition, the functional analysis of proteins elevated in BAT vesicles, in the FunRich database, showed the exosomal nature of these EVs. In parallel, a Reactome analysis confirmed the results of the functional analysis in the FunRich database, in which it was observed that EVs released by BAT from lean animals had a higher ratio of upregulated proteins related to mitochondria, lipid metabolism, electron transport chain, and the beta-oxidation pathway such as ACLY (ATP-citrate synthase), ACC (acetyl-CoA carboxylase 1), CPT2 (carnitine O-palmitoyl transferase 2, mitochondrial), and ACADM (medium-chain specific acyl-CoA dehydrogenase, mitochondrial) (Figure 5D and Figure 6B, Appendix A). In turn, EVs secreted by BAT from obese animals presented a higher ratio of upregulated proteins related to signal transduction, cell communication, inflammation, and obesity, such as annexin A6, MIF (macrophage migration inhibitory factor), CD14 (monocyte differentiation antigen), and ceruloplasmin (Figure 5C,D and Figure 6A, Appendix A).

A representative proteome map of the proteins found to be upregulated in obese and lean BAT-shed vesicles was established to complete our previous reports, which described characteristic protein maps of obese and lean visceral and subcutaneous secreted EVs (Figure 6) [9].

It can be concluded that proteome contents of white and brown adipose tissue shed-EVs show depot-specific protein profiles that reflect metabolic deregulation in animals with high fat diet intervention. More precisely, the EVs secreted by brown adipose tissue show protein loads that dynamically reflect changes in the cell of origin carrying biomarkers of mitochondrial BAT activity, thermogenesis, inflammation, and oxidative stress during the development of obesity.

### 2.3. Extracellular Vesicles Secreted by Brown Adipose Tissue Carry UCP1, Mitochondrial Components and Enzymes, and Proteins Associated with Obesity-Related Deregulated Metabolism in Obese Animals

Normalized DIA-SWATH/MS area values were used for a semi-quantitative analysis of selected proteins from the quantitative proteomic study, paying special attention to the proteins identified in BAT vesicles (Appendix A). The results from this analysis show a striking elevation of UCP1 in the EVs secreted by obese BAT vesicles as compared with lean BAT vesicles, and as compared with all the EVs secreted by white adipose tissues at both lean and obese states; this finding was also paralleled by FABP4 protein (Appendix A). Moreover, a significant elevation of this protein was detected in obese VAT vesicles as compared with lean VAT vesicles. In relation to FAS, there was statistically elevated significance in BAT EVs as compared with EVs released from white fat depots (Appendix A). MIF protein was also identified and analyzed, it was found to be upregulated in obese BAT vesicles as compared with lean BAT vesicles and all white vesicles, independently of the metabolic status (Appendix A). In the case of ceruloplasmin, which was always present in vesicles associated with obesity in the present work and in our previous analysis with human obese AT explants and murine obesity cell models [7,9], we found that it was significantly elevated in obese BAT EVs as compared with lean BAT EVs; however, the levels were similar to the vesicles secreted by the control and obese VAT vesicles; the lean and obese SAT vesicles showed the highest levels of ceruloplasmin with no differences among them (Appendix A). In relation to ACLY, there was a significant reduction in the obese BAT vesicles as compared with lean BAT vesicles, which in turn, showed elevated levels as compared with all white vesicles (Appendix A). It is also noteworthy to highlight the significant elevation of vinculin protein in obese VAT vesicles as compared with BAT vesicles; a difference also detected in obese VAT EVs as compared with lean VAT EVs (Appendix A). It is also interesting to mention that both lean and obese BAT vesicles were characterized by the elevated presence of complement 3 and perilipin 1 proteins (Appendix A).

In addition to a proteomic analysis, we isolated EVs from VAT, SAT, and BAT independent lean and obese animals to validate the SWATH results by immunodetection, focusing on BAT vesicles (*n =* 4 animals). Based on our previous experience and the available bibliography, we selected a panel of tissue-specific proteins and potential obesity/altered thermogenesis biomarkers [24,25]. From the identified proteins upregulated in BAT EVs, confirming the proteomic analysis, we observed a striking presence of FAS in both lean and obese BAT vesicles (Figure 7A,B), and ACLY, in the vesicles secreted by lean BAT tissue as compared with obese BAT tissue and all the other EVs at both lean and obese state (Figure 7A,C). In addition, elevated FABP4 was detected in obese BAT EVS as compared with lean BAT vesicles, and in all WAT vesicles independently of the metabolic status (Figure 7A,J). In the case of UCP1, we found a similar pattern, detecting a high and significant presence of this protein in the vesicles secreted by BAT tissue, and more importantly, in the vesicles secreted by BAT tissue from obese animals (Figure 7A,H). Perilipin-1 and caveolin-1 also showed a tendency to be elevated in the vesicles from BAT, especially, in the vesicles secreted by BAT tissue from obese animals; however, there was no statistical significance (Figure 7A,E,I). Ceruloplasmin showed no differences (Figure 7D). Notably, vimentin protein was found to be elevated in EVs from obese VAT as compared with EVs from other fat depots of both obese and lean animals (Figure 7A,F). Furthermore, CD14 protein was found to be statistically significantly elevated in the EVs of the SAT from lean animals as compared with the rest of the vesicles (Figure 7A,G). In addition, we observed, by immunodetection, that the vesicular proteins CD81 and Alix were found to be decreased in both obese and lean BAT EVs as compared with white fat EVs (Figure 7A).

In summary, the validation analysis confirms a characteristic elevation of vimentin protein in the vesicles shed by obese VAT, and increased CD14 protein in vesicles liberated by lean SAT. Moreover, it is revealed that EVs secreted by brown adipose tissue carry UCP1, which, together with FAS, FABP4, perilipin-1, and caveolin-1, is especially elevated in obese animals.

## 3. Discussion

In the present study, we isolated and characterized vesicles shed by whole explants from subcutaneous adipose tissue (SAT) and visceral adipose tissue (VAT), and, for the first time to the best of our knowledge, those secreted by brown adipose tissue (BAT) EVs or “batosomes”, of lean and obese animals. The proteomic analysis of these vesicles shows that, similar to our previous studies [7,9], protein composition is dynamic, being depot- and metabolic-status dependent. We show that AT EVs carry extracellular vesicle-structural components, as well as adipose tissue, obesity, and browning biomarkers, which vary regarding nutritional interventions. Additionally, we performed a quantitative label-free proteomic analysis of white and brown adipose tissue-secreted vesicles to reveal the proteins differentially present in EVs that are characteristic of each type of fat. Moreover, the proteins were validated by immunodetection in independent vesicles released by AT of lean and obese animals. Overall, the findings regarding EVs secreted by white adipose depots under high fat intervention in the established rat model, revealed new EV-associated proteins, and confirmed our previous findings, showing that visceral AT-secreted vesicles were more affected during the development of obesity than those from subcutaneous, thus, carrying interesting metabolic status and obesity-associated biomarkers related to central adiposity, inflammation, or insulin resistance. More interestingly, we report the first representative proteome map of BAT EVs (batosomes) that sets a starting point for future studies. Precisely, we reveal that brown adipose tissue of lean animals secretes EVs containing proteins associated with metabolism (beta-oxidation enriched), energy pathways, and cell growth, and that their protein cargo profiles change with HFD intervention. We demonstrate that, during the development of obesity, batosomes are enriched with proteins involved in signal transduction, cell communication, the immune response, inflammation, thermogenesis, and as potential obesity biomarkers. Therefore, this improved knowledge about the protein content of EVs, especially in the vesicles released by the BAT, could provide useful information for metabolic monitoring of obesity, and the efficacy of treatments, thus, opening a new paradigm in relation to metabolic diseases to identify pharmacological targets and innovative treatments.

Although extracellular vesicles have recently emerged as new mediators of physiological and metabolic cellular communication, which has generated great expectations [15], very little is known about vesicles secreted by adipose tissues or their compositions, especially concerning brown adipose tissue. However, recent research has begun to shed light on this question by describing circulating EVs as biomarkers of lipid and glucose metabolism in humans [26], or by studying vesicles released from human and murine adipose tissue explants and cell lines [16,18]. Thus, previous work with vesicles isolated from the primary culture of AT isolated from patients with SGBS (Simpson and Behmel syndrome) showed a reciprocal proinflammatory stimulation between adipocytes and macrophages, with the potential to aggravate local and systemic insulin resistance [16]. In the same direction, Eguchi et al. recently suggested that microvesicles released by cultured murine and human adipocytes, especially under hypertrophic conditions, had a promoting role in macrophage migration [27]. Thus, our own previous work has shown that EVs secreted by palmitate/oleic hypertrophied adipocytes, and those from insulin resistant, were able to stimulate healthy adipocytes differentiation and hypertrophy, and to induce insulin resistance; plus, those pathological vesicles promoted macrophage inflammation in vitro [7]. In addition, EVs from TAS explants from selected patients (with a mean BMI of 25.8 kg/m^2^) have been shown to cause alterations in insulin signaling of HepG2 hepatocytes [19]. Recently, it has also been shown that murine adipocyte-derived EVs could regulate POMC expression through hypothalamic mTOR signaling in vivo and in vitro, affecting body energy intake [28]. In addition, different investigations have studied the gene composition (microRNAs and mRNAs) of adipocyte-derived EVs, as they may exert post-transcriptional regulation in target cells/tissues at physiological and pathological levels [29,30]. For example, the study by Ogawa et al. identified 7000 mRNAs in EVs isolated from murine 3T3-L1 adipocytes, and these EVs were able to transfer these specific sequences to target cells such as macrophages [31]. Therefore, considering this functional evidence, we believe that it is necessary to describe the protein composition of EVs released by adipocytes or adipose tissue in the context of obesity. For that purpose, our research group has already made significant advances in deciphering the protein cargo of adipose tissue EVs by describing, for the first time, the proteome reference map of human obese and subcutaneous secreted EVs [9], and the functional and protein characterization of EVs secreted by adipocyte cell models of lipid hypertrophy and insulin resistance [7].

In this study, we describe, for the first time to best of our knowledge, not only the protein content of EVs liberated by white adipose tissue and its alteration with HFD intervention, but also the brown adipose tissue secreted vesicles as part of the batosomes [21], with the aim to shed light regarding the role of this thermogenesis specialized tissue in obesity. Brown adipose tissue (BAT) is the primary site of no shivering thermogenesis, and for that reason, it plays an important role in adaptive energy expenditure processes [32]. Brown adipocytes are greatly enriched in mitochondria that contain uncoupling protein-1 (UCP1) in their inner membrane, where it uncouples the respiratory chain from oxidative phosphorylation; this permits brown adipocytes to actively oxidize substrates to produce heat. Different studies have demonstrated that this process is sensitive a cold environments and also to diet through diet-induced thermogenesis [33]. BAT-mediated thermogenesis can hence protection against obesity via boosting energy expenditure [34].

Previous reports have described the characterization of the proteome of the different subtypes of extracellular vesicles secreted by murine 3T3-L1 adipocyte cells, and also the proteome of the exosomes secreted by the murine 3T3-F442A cell line [12,13]. A previous study by our group characterized, for the first time, the protein composition of EVs secreted by pathological murine C3H10T1/2 adipocytes with insulin resistance and/or lipid hypertrophy, by combining treatments with high-dose glucose and insulin or with fatty acids; thus, observing different protein charges, and therefore, possible biomarkers depending on the cell stimulus of origin [7]. These studies yielded relevant information about the composition, but, as they were performed on adipocyte cell lines, they lacked information about the anatomical location, and about the whole physiology of adipose tissue, omitting those vesicles secreted by other cellular constituents, their interaction, and their relationship with the extracellular matrix (ECM). In this context, a study by Kranendonk et al. showed the proteome of EVs released by human subcutaneous preadipocytes differentiated in vitro to adipocytes from lean to moderately obese women [16]. In addition, our group has recently characterized the protein cargo of EVs secreted by whole tissue visceral and subcutaneous fat from morbidly obese patients, and found biomarkers of tissue type according to anatomical location, central adiposity, inflammation, and obesity-associated diseases [9]. However, there is little information on the EVs secreted by brown adipose tissue in the context of obesity. Recent studies have shown that exosomes from brown adipose tissue could communicate with other metabolic organs, thus mitigating metabolic syndrome [24]. Therefore, characterization of the protein cargo of vesicles released by BAT will allow us to discern the molecular events responsible for their metabolic benefits and elucidate their potential as biomarkers of BAT activity.

In the present study, we isolated and characterized EVs from adipose tissue explants of animals with diet-induced obesity (DIO) and lean animals (standard diet). We observed differences in the secretory profiles of adipose tissue depending on the anatomical location and metabolic status; visceral adipose tissue secreted a higher proportion of EVs than subcutaneous adipose tissue, thus this was in agreement with our previous studies [9]. Interestingly, adipose tissue from lean animals was also observed to secrete a greater number of EVs than fat depots from obese animals. These results were unexpected, as the presence of stress or pathology, including obesity, has been shown to induce an increase in circulating vesicles [35]. However, our own previous studies have demonstrated a decrease in particle secretion measured by a NTA nanoparticle tracking analysis (NTA) in murine adipocytes with insulin resistance and lipid hypertrophy as compared with healthy adipocytes [7]. However, we demonstrated by different techniques that EVs secreted by white and brown adipose tissue from both obese- and normal-weight animals were of similar size regardless of cellular origin or metabolic state, in agreement with our previous experience [7,9]. It should be noted that depending on the technique, NTA or ExoView, we observed different particle sizes and concentrations for the same tissue. These differences may be due to the fact that the EVs analyzed by ExoView are a selected population of particles captured by classical small/endosomal vesicle markers (tetraspanins), and that interferometry allows the analysis of particles with a minimum size of 50 nm, unlike NTA, which detects heterogeneous populations of particles larger than 100 nm [36]. Furthermore, it should also be noted that EVs released by AT have a profile of small vesicle and endosomal markers, such as the tetraspanins CD9, CD63, CD81, a profile, importantly, that varies according to the type of tissue or metabolic state. This agrees with the study by Mizenko et al., in which it was observed that the profile of tetraspanins varied according to the type of sample analyzed, the development of the pathology, and the isolation technique [37]. Therefore, it can be concluded that there is a specific EV tetraspanin profile that depends on the location of the fat and varies with the metabolic deregulation of the tissue.

As far we are aware, this is the first proteome characterization of the EVs secreted by the BAT of obese and lean rats according to the anatomical location of the tissue and their nutritional status. Recent studies have characterized the proteomic profile of murine brown adipose tissue secretome [38,39] and of primary culture of adipocytes from brown adipose tissue of the supraclavicular region [40]. The study by Deshmukh et al. showed that certain identified proteins had a vesicular origin, but they did not analyze the composition of these EVs in addition to offering a view only of the adipocyte itself and not of the joint relationship of the adipose tissue, nor of the influence of the nutritional condition [40], unlike our analysis. In addition, there have been recent studies on the role of microRNAs secreted by BAT. MiR-99b has been postulated as a biomarker of BAT activity, since it has been observed that the EVs released by the activated BAT containing this microRNA repressed the expression of Fgf21 at the hepatic level [24]. In addition, the study by Kariba et al. showed that miR-132-3p of EVs released by BAT under stress conditions attenuated the expression of lipogenic genes by suppressing the expression of Srebf1 in the liver [41]. Interestingly, in our study, both obese and lean BAT EVs had a significant mitochondrial component (40–45%), unlike EVs shed by white adipose tissue. It was observed that the brown vesicles contained proteins related to the mitochondrial metabolism of the brown tissue and to energy pathways. For the first time, we have identified the BAT thermogenesis marker, UCP1 (brown fat mitochondrial uncoupling protein 1), in EVs of brown origin. Furthermore, it should be noted that this protein, i.e., UCP1, was found to be elevated in EVs from BAT as compared with EVs from WAT (subcutaneous and visceral), and curiously upregulated in EVs from brown tissue of obese animals as compared with EVs from the same tissue in lean animals. This result does not seem to agree with the literature, since, although BAT has a greater resistance to inflammation than white adipose tissue, it is known that, in obesity, the secretion of proinflammatory cytokines seems to alter the thermogenic function of BAT with a reduction in the expression levels of UCP1 and other markers of thermogenesis [42]. In murine models of genetic obesity, a decrease in BAT activation has been reported, probably as a consequence of an association with the reduction of UCP1 [43]. However, in diet-induced obesity (DIO) models, some studies have indicated that UCP1 levels remained stable or, as in the work by Alcalá M. et al., UCP1 protein levels increased in obese mice after HFD intervention for 20 weeks, which was not comparable to our study of just 9 weeks, but its coherence showed an evolution [43,44]. Therefore, we hypothesize that BAT EVs reflect the adaptive state of this tissue which triggers a diet-induced adaptive thermogenic response, leading to increased protein levels of UCP1. However, it remains to be confirmed whether this is a time-sensitive effect, since, in this study, the HFD was maintained for nine weeks, unlike in previous studies [44]. In addition, our results showed that VAT EVs from obese animals contained a higher ratio of UCP1 as compared with SAT EVs from normal-weight and obese animals. This result is consistent with other reports, in which a higher level of UCP1 gene expression was observed in VAT from obese patients as compared with SAT from lean and obese patients [45]. Moreover, among the proteins differentially regulated in the EVs of obese BAT, it is interesting to highlight enzymes, cytokines, and proteins related to energy and insulin signaling pathways such as C3/C4, MIF1 (macrophage migration inhibitory factor), REEP5 (receptor expression-enhancing protein 5), PEPCK-C (phosphoenolpyruvate carboxykinase), AOC3 (primary membrane amine oxidase), TARG1 (regulator of GLUT4 trafficking 1), and Gyk (glycerol kinase). These proteins have also been found in the murine and human secretome of BAT [25]. A protein of the same family as REEP5, i.e., REEP6, has been shown to be regulated by thermogenic stimuli in adipose tissue, and furthermore, REEP6 KO mice were defective in cold-induced thermogenesis, showing an obesity-prone phenotype [46]. Additionally, overexpression of PEPCK-C, a regulatory enzyme of glyceroneogenesis in adipose tissue, has been described to lead to a high susceptibility to diet-induced insulin resistance and obesity [47]. Moreover, in this context of obesity and insulin resistance, numerous reports have evidenced increased circulating AOC3 in diabetic conditions; thus, it was described as upregulated in the adipose tissue of obese/diabetic rodents and especially in the hypertrophic fat pads [48]. Furthermore, TARG1 colocalizes with GLUT4 and positively regulates GLUT4 trafficking and insulin sensitivity in adipocytes; in this regard, within the insulin signaling pathway, it is postulated that TARG1 is a substrate of GSK3 and its phosphorylation is regulated through the PI3K/AKT/GSK3 axis [49]. In addition, findings have suggested that Gyk, an important gene in the browning process of white adipocytes, stimulated UCP1 expression through a mechanism that was partially dependent on the βAR-cAMP-CREB pathway and Gyk-mediated regulation of fatty acid metabolism [50]. Notably, glycerol kinase activity per adipose cell was elevated in all obese rodents relative to lean controls, and was generally correlated with adipose cell size [51]. In rats and mice, BAT has been shown to possess much higher glycerol kinase activity than white adipose tissue (WAT) [51], paralleling our findings on obese BAT-EVs. It should be mentioned, herein, that 24 different proteins identified in the BAT EVs proteome have been previously described in the BATLAS [23] database such as UCP1, Gyk, basigin, and many mitochondrial enzymes, confirming that we have accurately isolated vesicles of brown adipose tissue.

In this study, we identified and observed that the proteins ACLY (ATP citrate synthetase), Rab 14, ACC (acetil-CoA carboxilase), and 2-enoyl thioester reductase were decreased in the EVs of the BAT of obese animals as compared with the same tissue of lean rats. ATP citrate synthase is an enzyme that participates in lipogenesis, and it is known that, during the development of obesity, there is a decrease in its activity [52]. Notably, the Rab 14 protein plays a major role in GLUT4 trafficking, thus, controlling the transit of internalized GLUT4 from early endosomes to the Golgi complex [53]. Furthermore, ACC and 2-enoyl thioester reductase, de novo lipogenesis anabolic genes, are known to be expressed under conditions of thermogenesis and to correlate with UCP1 expression [54]. Therefore, considering the findings in our analysis, there is a need to know if the cytosolic proteins and enzymes that travel inside EVs are functional once they reach a target cell; nevertheless, we can postulate that EVs secreted by BAT might be a good source of biomarkers of metabolic status and of thermogenesis activity.

Furthermore, in the context of obesity, among the proteins common to all EVs analyzed in the present work, we found similarities to those characterized in our previous studies [7,9]; thus, we identified structural and cytoskeletal proteins such as annexins A1/A6 that modulated anti-inflammatory processes and could be altered in obesity-associated type 2 diabetes [55] and histones such as H4 of which epigenetic changes such as lysine acetylation are known to trigger metabolic disorders in mice with diet-induced obesity (DIO) [56]. In addition, it is interesting to note that proteins related to fatty acid synthesis or transport or adiposity were identified in all the analyzed EVs, such as caveolin-1 (cav-1), fatty acid synthetase (FAS), adipocyte fatty acid binding protein (FABP4), and perilipins (perilipin-1). It should be noted that caveolin-1 and perilipin-1 proteins had already been proposed in the literature as vesicular biomarkers of adipose tissue [57], especially perilipin-1, which was identified as a biomarker to detect adipocyte-derived EVs in the circulation [58]. Thus, we found that the perilipin-1 protein was observed in a higher ratio in EVs released from fat depots under obese conditions, especially from visceral and brown fat. This result was consistent with previous reports describing that perilipin-1 was significantly increased at the circulating level in C57BL mice with diet-induced obesity [27]. In addition, caveolin-1 protein was found to be elevated in BAT EVs from obese rats as compared with lean counterparts; and it was generally found to be upregulated in EVs shed from the different obese depots. EVs carrying this protein are involved in communication between different cell types within adipose tissue, specifically between endothelial cells and adipocytes [57]. Recent studies have also shown that this protein was related to metabolic alterations and could influence tumor development or progression by controlling metabolism through glycolysis, fatty acid metabolism, or mitochondrial pathways [59]. In addition, the FABP4 protein was found to be elevated in EVs released from BAT and VAT under obese conditions as compared with EVs released from subcutaneous fat depots and under control-weight conditions. Increased circulating levels of this protein, i.e., FABP4, are known to correlate with the incidence of metabolic diseases, and reduced levels are associated with improved metabolic health [60]. FAS protein is another protein that has been found in EVs secreted by adipose tissue in animals with obesity. It is known that this protein may contribute to impaired insulin sensitivity and AT dysfunction in obesity [61].

Consequently, our study has led to the identification of new adipokines in AT vesicles from obese animals that are also of interest, including proteins characterized in EVs in our previous studies [7,9], such as DDP2, ceruloplasmin, CD14, or vimentin. The presence of the transmembrane glycoprotein DDP2 in obese BAT vesicles is especially significant. Noteworthy, proteins from this family are known to play an important role as adipokines, specifically DDP4, which has been suggested as a biomarker of visceral obesity, insulin resistance, and metabolic syndrome [62]. Moreover, we found ceruloplasmin protein to be elevated in EVs released from adipose tissue under conditions of obesity and lipid hypertrophy, especially in visceral and brown fat depots. It is noteworthy that this protein has been previously postulated as a biomarker of obesity, since a significant increase in the circulating level of this protein has been observed in individuals with obesity as compared with lean individuals [63]. Therefore, we hypothesize that ceruloplasmin is secreted in EVs in a higher proportion during the development of obesity. We also observed that CD14 protein and vimentin were in a higher ratio in EVs from subcutaneous and visceral fat depots, respectively. CD14 protein, also elevated in obese BAT vesicles, is related to macrophage infiltration in the AT and causes an increased inflammatory state of this tissue [64]. Vimentin protein plays a key role in AT plasticity, and has also been found to be involved in obesity and diabetes [65].

To sum up, we believe that we have described, for the first time, both qualitatively and quantitatively, the protein load of the EVs shed from the AT (white and brown) of rats according to the type and anatomical location of this tissue. This study is an important starting point for a better understanding of this alternative form of communication in the context of metabolic diseases. In addition, the knowledge of these proteins can be very valuable as biomarkers of diseases, since we have shown that the content of EVs varies depending on the type and anatomical location of the fat depots (SAT, VAT, and BAT) and the BMI of the animal (obese or lean). Thus, it should be noted that we have observed that EVs of brown origin have different protein loads depending on the metabolic state and that they dynamically reflect changes in the cell of origin carrying biomarkers of mitochondrial BAT activity, thermogenesis, inflammation, and oxidative stress during the development of obesity.

## 4. Materials and Methods

### 4.1. Animal Lean and DIO Model

Forty adult Sprague–Dawley male rats weighing 150–199 g and 5 weeks of age were acquired from the animal facility of the University of Santiago under the procedure with code 15005/2015/003, according to the institutional guidelines and the European Union standards for the care and use of experimental animals. The procedures were approved by the Consellería de Medio Rural, Xunta de Galicia, and the Animal Care Committee of Santiago de Compostela University (Santiago de Compostela, Spain). The animals were kept for 7 days in the animal husbandry to enable acclimatization to the local conditions (temperature (22–24 °C) and light/dark cycle (12 h light, 12 h darkness)). After acclimation, the animals were randomly separated into lean and DIO obese experimental groups (*n*  =  20/per group).

Precisely, DIO animals were fed with 60% Kcal fat, 20% Kcal protein, and 20% Kcal carbohydrate, which resulted in 5.21 kcal/g (D12492, Research Diets, NJ, USA). The age- and sex-matched control rats were fed with a standard (low fat diet (LFD)) diet with 10% fat content (D12450B, Research Diets, NJ, USA) with an energy density of 3.82 kcal/g (0% Kcal fat, 20% Kcal protein, and 70% Kcal carbohydrate). The experimental groups were both fed ad libitum for 9 weeks; food intake and body weight were measured weekly. Moreover, fasting glucose was measured by tail incision 1 week before the euthanasia using Accu-check Performa device (Roche, Basel, Switzerland).

### 4.2. Rat Adipose Tissue Explants Culture

Visceral (VAT), subcutaneous (SAT), and brown (BAT) adipose tissue were obtained from 13 lean and 13 obese animals and excised in 1 g pieces for culturing. Visceral fat (VAT) was excised from the hypogastric region around the internal organs, subcutaneous fat (SAT) from the inguinal region, and brown fat (BAT) from the interscapular region. Tissues were transported to the laboratory in sterile KRH buffer with penicillin (100 U/mL) and streptomycin (100 μg/mL) for further processing.

### 4.3. Secretome Collection

Adipose tissue samples (*n* = 78: 26 SAT, 26 VAT and 26 BAT) were processed as previously described with some modifications to obtain tissue secretomes [9,66]. In brief, whole adipose tissue explants were processed to remove any contaminants by doing an intensive wash in PBS that was repeated several times. The tissue pieces were transferred to a tube containing 25 mL of PBS and centrifuged for 5 min, at 1800 rpm, at room temperature to eliminate red blood cells and debris. 1 g pieces of each tissue type (VAT, SAT, and BAT) from independent animals were incubated at 37 °C and 5% CO_2_ in 5 mL/tissue piece/well with serum-free DMEM medium without phenol red (Sigma-Aldrich, Burlington, MA, USA) supplemented with 1% (*v*/*v*) penicillin-streptomicin. The medium was changed after 2 and 24 h. After the last wash (time point 24 h), all dishes received fresh DMEM medium (3 mL/1 g tissue) and were left in culture for an additional 48 h to allow the secretion of vesicles. Then, the media were collected, centrifuged during 5 min at 1800 rpms, and stored at −80 °C for further analysis.

### 4.4. Isolation of EVs by Differential Ultracentrifugation

The adipose tissue EVs were isolated from secretomes following our previous well established protocols [7,9]. The secretomes were filtered with a 0.22 µm filter to eliminate contaminating cell debris and centrifuged in a Beckman Coulter Optima XPN-100 Ultracentrifuge at 10,000× *g* at 4 °C for 20 min, followed by 10,000× *g* ultracentrifugation at 4 °C for 90 min with a Type SW 41 Ti rotor, acceleration and deceleration brake profile 9, to pellet EVs following previous reports [67,68]. The supernatant was carefully separated, and vesicle-containing pellets were resuspended in ice-cold PBS. A second round of ultracentrifugation (10,000× *g* at 4 °C for 90 min) was performed, and the resulting vesicle pellet was resuspended in PBS or RIPA sample buffer according to the analysis.

### 4.5. Immunoblotting

EV pellets secreted by 1 g of VAT, SAT, and BAT explants were lysed, resuspended in *Laemmli* sample buffer, and processed for immunoblotting, as previously described (Camino T, Lago-Baameiro N; Bravo SB, Molares-Vila A, Sueiro A, Couto I, Baltar J, Casanueva FF, n.d.) [6]. Primary anti-Alix (sc-53538, dilution 1:500), anti-CD81 (sc-166029, dilution 1:500), anti-ceruloplasmin (sc-365205, dilution 1:500), anti-fatty acid synthase (sc-48357, dilution 1:500), and anti-CD14 (sc-515785, dilution 1:500) were purchased from Sta. Cruz Biotechnology (Santa Cruz, CA, USA); anti-Perilipin1 (PA5-55046, dilution 1:1000), anti-Vimentin (PA5-27231, dilution 1:1000), and anti-FABP4 (PA5-19811, dilution 1:1000) were purchased from ThermoFisher Scientific S.L (Waltham, MA, USA); anti-ACLY (4332s) was purchased from Cell Signaling Technology (Danvers, MA, USA); and anti-UCP1 (ab10983) was purchased from Abcam (Cambridge, UK).

### 4.6. Nanoparticle Tracking Analysis (NTA)

Vesicle size and concentration distribution were analyzed using nanoparticle tracking analysis (NTA) in a Malvern NanoSight NS300 equipment NTA version NTA3.3 Dev Build 3.3.104 (v3.3; Panalytical, Ltd., Malvern, UK), following the manufacturer’s instructions with the following settings: camera level 14, slider shutter 1256, slider gain 366, syringe pump speed 40, and detect threshold 4.

### 4.7. ExoView Analysis

Complete characterization of secretome-derived EVs (*n* = 24: 4 SAT, 4 VAT and 4 BAT for each group (obese and lean animals)) was performed by ExoView R100 (Unchained labs; Pleasanton, California, USA) using mouse tetraspanins kits (EV-TETRA-M2). Chips (*n* = 30) were prescanned using the provided protocol to identify any previously adhered particles during manufacturing. For incubation, chips were placed in plates of 12 wells avoiding contact of the chip corners with the sides of the well. Secretomes were diluted 1:25 in the provided incubation solution buffer; 50 μL of the diluted sample was incubated overnight without agitation. Following the manufacturer´s protocol, several washes were performed the following day. Then, 1 µg/mL of fluorescently labeled antibodies provided by the kit (anti-CD9 kit (CF 488A), anti-CD81 (CF 555A), and anti-CD63 (CF 647A)) or against our own proteins of interest (anti-UCP1 (CF 647A), anti-FABP4 (CF 594A), and anti-caveolin-1 (CF 488A) labeled by Alexa Fluor Conjugation Kits Fast (ABCAM)) were incubated for one hour with gentle agitation. Then, the chips were washed and dried for analysis using an ExoView Analyzer (Unchained labs; Pleasanton, California, USA).

### 4.8. Mass Spectrometric DDA Qualitative Analysis and Protein Quantification by DIA-SWATH (Sequential Window Acquisition of All Theoretical Mass Spectra)

To perform qualitative and quantitative identification of the EVs’ cargo proteins, we processed vesicles obtained from 1 g of each type of adipose tissue (SAT, VAT, and BAT) from 8 independent animals (4 lean and 4 obese). Digested peptides were separated by reverse-phase chromatography. The gradient was created using a microfluidic chromatography system (Eksigent Technologies nanoLC 400, SCIEX, Foster City, CA, USA) coupled to a Triple TOF 6600 high-speed mass spectrometer (SCIEX) with a microflow source using a data dependent acquisition (DDA) method. For this analysis, only proteins with FDR < 1% (99% protein confidence) were selected. For relative quantification using the DIA-SWATH method, first, we built a spectral library that grouped each condition group into a pool. Then, equal amounts of each sample type (*n* = 24) were run in the TripleTOF 6600 using a SWATH-MS acquisition method.

### 4.9. Protein Functional Analysis

The functional analysis was performed using the FunRich open access software (Functional Enrichment analysis tool version 3.1.3) for functional enrichment and interaction network analysis (accessed on 1 July 2022) [69], the Panther online Classification System (accessed on 1 July 2022) [70], and Reactome online pathway database (accessed on 27 July 2022) [71].

### 4.10. Statistical Analysis

To achieve the global qualitative proteomic analysis, the data files were processed using the ProteinPilot^TM^ 5.0.1 software (SCIEX) which uses the algorithm Paragon^TM^ for database searching and Progroup^TM^ for data grouping. Data were searched using a RAT-specific Uniprot database). The false discovery rate was performed using a nonlinear fitting method that displayed only those results that reported a 1% global false discovery rate or better [72].

To make the SWATH quantitative analysis, the targeted data extraction of the fragment ion chromatogram traces was performed by PeakView (version 2.2) using the SWATH Acquisition MicroApp (version 2.0). For the PCA and cluster analysis, the R 3.5.3 version and “base”, “stats”, “gplots”, “Hmisc”, and “car” packages were used after normalization of raw data. The principal component analysis (PCA) was applied, based on the correlation matrix, due to its pairwise two-sided *p*-values that were 0 for the entire matrix, and therefore, all of them were statistically significant, which could explain the good success in the application of PCA. Therefore, 87.70% of the total variability of the data could be explained by the two first principal components. For cluster analysis, unpaired Student’s t-test for mean comparisons between each and differentiated samples were previously applied, and used Euclidean distance, suitable for quantitative variables and complete linkage as cluster criteria. For differentially expressed protein selection, a fold change ≥1.5 and a *p*-value ≤ 0.05 were selected.

The statistical significance among multiple groups was analyzed by one-way ANOVA and Kruskal–Wallis test followed by Dunn’s multiple comparison test in GraphPad Prism 6 software; *p* ≤ 0.05 was considered to be statistically significant.

## 5. Conclusions

Brown adipose tissue secretes EVs or batosomes of exosomal signature (CD81/CD9/Alix positive) with a smaller size than the vesicles released by white adipose tissues (subcutaneous and visceral).

A DDA proteomic analysis identified 100 proteins unique to and characteristic of vesicles from obese BAT tissue and 20 from lean BAT tissue. Sixty percent of the proteins were unique to BAT vesicles as compared with those of white adipose tissues (VAT and SAT), most of them being components of mitochondria including unique BAT characteristic UCP1 protein.

A comparative DIA-SWATH quantitative proteomic analysis showed 88 proteins elevated in the vesicles of obese BAT tissue as compared with lean BAT tissue, and among these, proteins such as UCP1, Glut1, MIF, ceruloplasmin showed a higher fold change.

Batosomes of lean animals contain proteins associated with mitochondria, lipid metabolism, electron transport chain, and the beta-oxidation pathway, and their protein cargo profiles are dramatically affected by high fat diet (HFD) intervention.

In obese animals, batosomes are enriched with proteins involved in signal transduction, cell communication, the immune response, inflammation, thermogenesis, and as obesity biomarkers including UCP1, Glut1, MIF, annexin A6, CD14, and ceruloplasmin.

Thus, the protein content of EVs varies depending on the type and anatomical location of the fat depots (SAT, VAT, or BAT) and the BMI of the animal (obese or lean). Precisely, batosomes have different protein loads depending on the metabolic status that dynamically reflects changes in the cell of origin carrying biomarkers of mitochondrial BAT activity, thermogenesis, inflammation, and oxidative stress during the development of obesity.

## Figures and Tables

**Figure 1 ijms-23-10826-f001:**
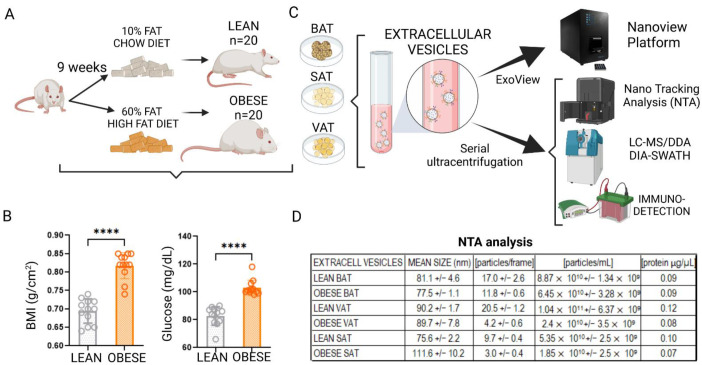
Establishment and characterization of a model of diet-induced obesity (DIO) for EV isolation. (**A**) A schematic of the animal model for obtaining adipose tissue from normal weight (control) or obese animals; (**B**) BMI (body mass index, g/cm^2^) and basal glucose (mg/dL) in control (LFD, low fat diet, *n* = 20) and obese (60% HFD, *n* = 20) rats after 9 weeks of diet intervention. EVs were isolated from 1 g of whole adipose tissue explants (SAT, subcutaneous adipose tissue; VAT, visceral adipose tissue; BAT, brown adipose tissue) by ultracentrifugation for nanoparticle tracking analysis (NTA), proteomics (LC-MS qualitative DDA-/quantitative DIA-SWATH), and immunodetection; (**C**) additionally, EVs were directly analyzed using a ExoView platform. Particle concentration/mL according to size (nm) of EVs released from SAT, VAT and BAT of control and obese rats obtained by NTA; (**D**) a summary of the averages together with the standard deviations of the size (nm), particle/frame concentration, particles/mL, and protein concentration of these vesicles isolated from the secretome of 1 g of AT. Differences were analyzed using a Mann–Whitney *U* test; *p* ≤ 0.05 was considered to be statistically significant (**** *p* < 0.0001) (created with BioRender.com).

**Figure 2 ijms-23-10826-f002:**
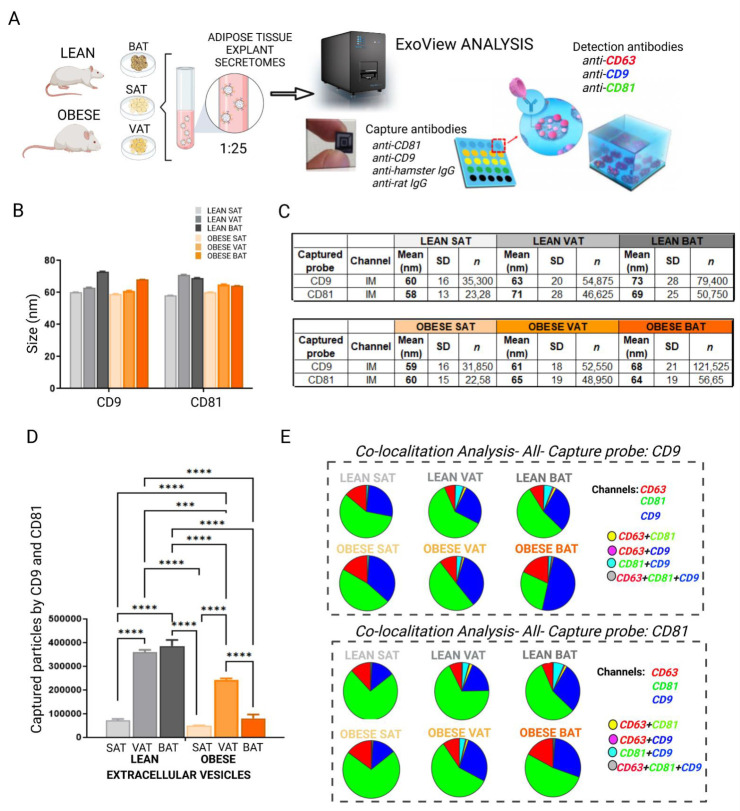
Complete characterization of EVs isolated from white, visceral (VAT), subcutaneous (SAT), and brown (BAT) adipose tissue explants from obese and normal-weight rats using the ExoView platform: (**A**) The ExoView technology allowed the analysis of the size, captured particle concentration, and co-localization of tetraspanins CD63, CD9, and CD81 of the different EVs isolated from white and brown adipose tissue explants from obese and control rats, by immunocapture with anti-CD9 and -CD81 on individualized chips. (**B**,**C**) the size of EVs was analyzed according to the CD9- or CD81-captured antibody by interferometry; (**D**) the number of particles captured by CD9 and CD81 tetraspanins; (**E**) co-localization of CD63 (red), CD81 (green), and CD9 (blue) tetraspanins on EVs was analyzed according to their capture by CD9 or CD81. At least 4 independent assays were performed for each condition, and differences were analyzed using a one-way ANOVA and Kruskal–Wallis test, *p* ≤ 0.05 was considered to be statistically significant (*** *p* < 0.001; **** *p* < 0.0001) (created with BioRender.com).

**Figure 3 ijms-23-10826-f003:**
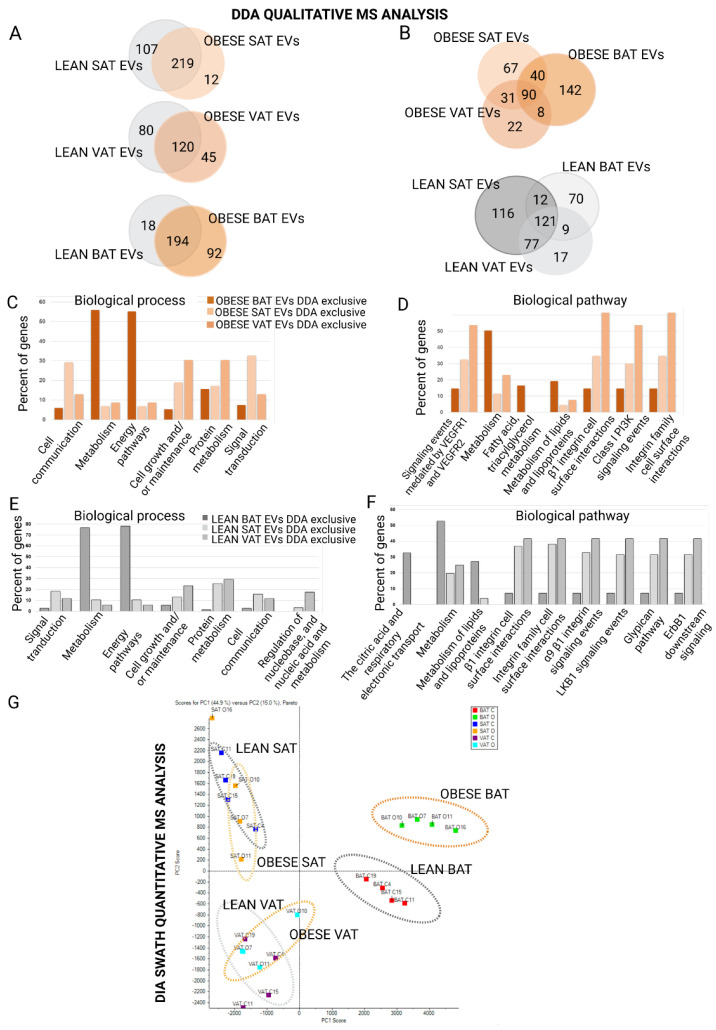
Qualitative (DDA) and quantitative (DIA-SWATH) analysis of the proteome of vesicles released from white (visceral, subcutaneous) and brown AT from lean (control) and obese animals: (**A**,**B**) Descriptive and comparative Venn diagram showing the total number of proteins identified with an FDR < 1% (99% protein confidence) in vesicles isolated from visceral (VAT), subcutaneous (SAT), and brown (BAT) adipose tissue from 4 control and 4 independent obese animals. Functional analysis (FunRich) classification of proteins identified by: (**C**,**E**) biological processes; (**D**,**F**) pathways, for vesicles isolated from adipose tissue of obese (**C**,**D**) and normal weight (**E**,**F**) animals; (**G**) PCA analysis of DIA-SWATH transformed areas for quantitative comparison of all samples *(n* = 24 (*n* = 8 rats, 4 lean and 4 obese; *n* = 8 SAT, *n* = 8 VAT, and *n* = 8 BAT; for 3 technical replicates for each tissue sample)). FDR, false discovery rate.

**Figure 4 ijms-23-10826-f004:**
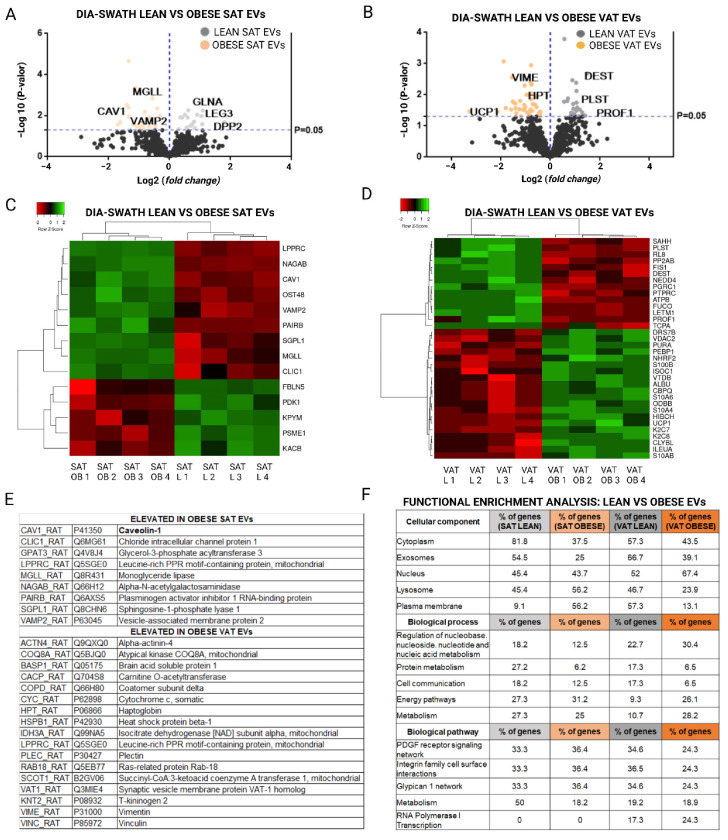
Quantitative DIA-SWATH/MS analysis of protein content in EVs secreted by subcutaneous adipose tissue (SAT) and visceral adipose tissue (VAT) in normal (control) and obese animals. (**A**,**B**) A volcano plot representation of elevated proteins in SAT EVs and VAT EVs from lean versus obese animals with a fold change >1.5; (**C**,**D**) cluster analysis by heat map representation of proteins in SAT EVs and VAT EVs with a fold change ≥2 and a *p*-value ≤ 0.05. (**E**) summary table with elevated proteins in EVs released from SAT and VAT of obese animals; (**F**) classification of the comparative functional enrichment (FunRich basis) according to cellular component, process, and biological pathway of the elevated proteins in SAT EVs and VAT EVs from normal (grey) and obese (orange) animals (**F**).

**Figure 5 ijms-23-10826-f005:**
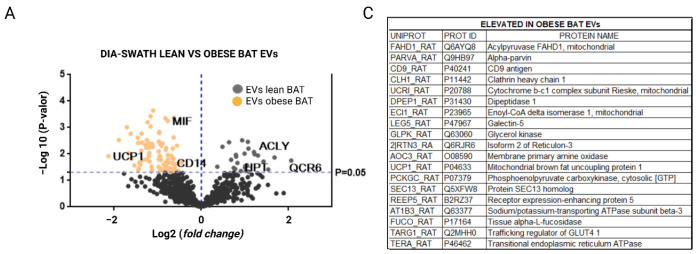
Quantitative DIA-SWATH/MS analysis of protein content in EVs secreted by brown adipose tissue (BAT) in normal (control) and obese animals. (**A**) A volcano plot representation of elevated proteins in BAT EVs from lean versus obese animals with a fold change >1.5; (**B**) cluster analysis by heat map representation of proteins in BAT EVs with a fold change ≥2 and *p*-values ≤ 0.05; (**C**) summary table with elevated proteins in EVs released from BAT of obese animals; (**D**) classification of the comparative functional enrichment (FunRich basis) according to cellular component, process, and biological pathway of the elevated proteins in BAT EVs from normal (grey) and obese (orange) animals.

**Figure 6 ijms-23-10826-f006:**
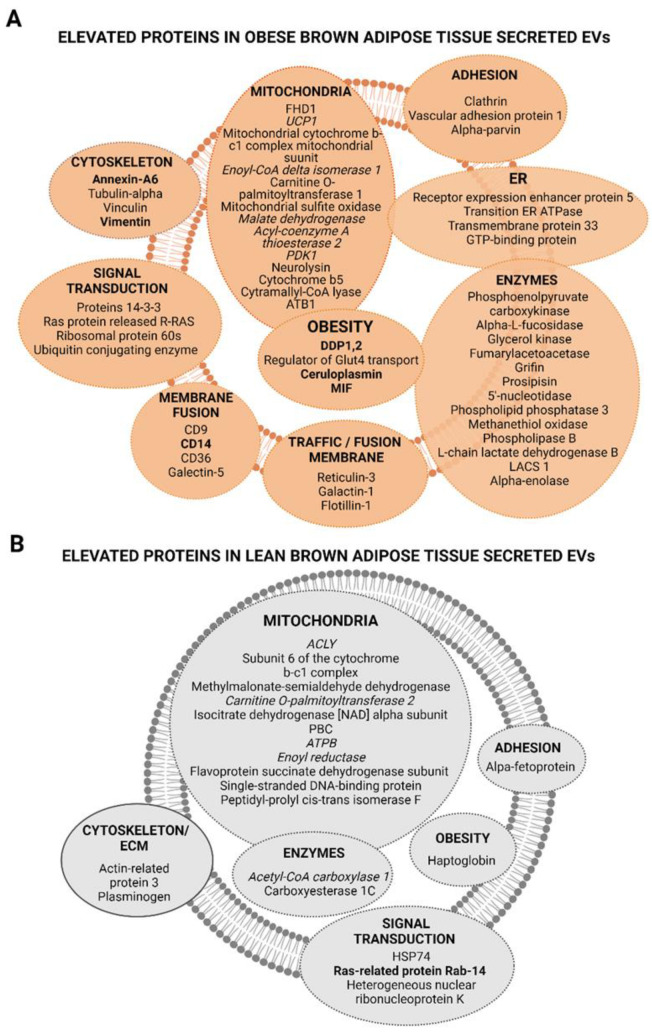
Proteome maps of upregulated proteins in brown adipose tissue (BAT) EVs (i.e., “batosomes”) from obese (**A**) and normal weight (**B**) animals with a fold change >1.5. Proteins in italics are related to beta-oxidation, and in bold to obesity and inflammation. ER, endoplasmic reticulum (Created with BioRender.com).

**Figure 7 ijms-23-10826-f007:**
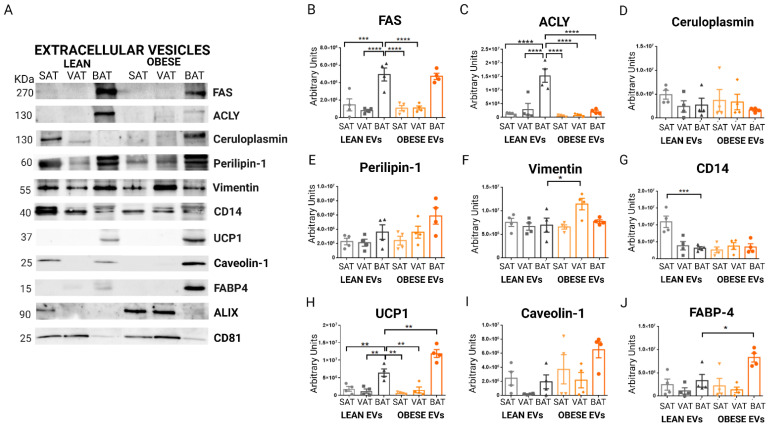
Immunodetection validation of vesicular proteins identified as potential biomarker candidates for brown adipose tissue (BAT) and white fat in the DIA-SWATH/MS analysis (sequential window acquisition of all theoretical fragment ions) in EVs secreted from 1 g of AT from obese and lean animals. Representative images are shown of: (**A**) immunodetection; (**B**) graphs of band quantification for FAS protein (fatty acid synthase); (**C**) ACLY (ATP-citrate synthase); (**D**) ceruloplasmin; (**E**) perilipin-1; (**F**), vimentin; (**G**) CD14; (**H**) UCP1 (mitochondrial uncoupling protein of brown fat 1); (**I**) caveolin-1; (**J**) FABP4 (fatty acid binding protein 4 of adipocytes) (*n* = 4 independent lysates for each sample type). Differences were analyzed using a one-way ANOVA and Kruskal–Wallis test, followed by Dunn’s multiple comparison test (*p* ≤ 0.05 was considered to be statistically significant; * *p* < 0.05, ** *p* < 0.01, *** *p* < 0.001, and **** *p* < 0.0001). Raw densitometry data, and complete immunoblot images are shown in Appendix A. EVs, extracellular vesicles; SAT, subcutaneous adipose tissue; VAT, visceral adipose tissue; BAT, brown adipose tissue.

**Table 1 ijms-23-10826-t001:** Identified proteins by quantitative DIA-SWATH ^1^ analysis in EVs secreted by visceral (VAT), subcutaneous (SAT), and brown adipose BAT) tissues of lean and obese animals.

		DIA-SWATH Analysis		
		Elevated in Lean SAT EVs		
UNIPROT	PROT ID	Protein Name	*p*-Value	Fold
KACB_RAT	P01835	Ig kappa chain C region, B allele	0.026	4.44
PDK1_RAT	Q63065	[Pyruvate dehydrogenase (acetyl-transferring)] kinase isozyme 1	0.021	3.50
PSME1_RAT	Q63797	Proteasome activator complex subunit 1	0.006	2.13
KPYM_RAT	P119802	Isoform M2 of pyruvate kinase PKM	0.026	2.11
FBLN5_RAT	Q9WVH8	Fibulin-5	0.011	2.07
		**Elevated in Obese SAT EVs**		
PAIRB_RAT	Q6AXS5	Plasminogen activator inhibitor 1 RNA-binding protein	0.027	3.24
CLIC1_RAT	Q6MG61	Chloride intracellular channel protein 1	0.021	3.06
SGPL1_RAT	Q8CHN6	Sphingosine-1-phosphate lyase 1	0.003	2.64
MGLL_RAT	Q8R431	Monoglyceride lipase	0.004	2.54
CAV1_RAT	P41350	Caveolin-1	0.011	2.53
LPPRC_RAT	Q5SGE0	Leucine-rich PPR motif-containing protein, mitochondrial	0.000	2.53
VAMP2_RAT	P63045	Vesicle-associated membrane protein 2	0.025	2.27
NAGAB_RAT	Q66H12	Alpha-N-acetylgalactosaminidase	0.000	2.18
OST48_RAT	Q641Y0	Dolichyl-diphosphooligosaccharide--protein glycosyltransferase 48 kDa sub	0.034	2.12
		**Elevated in Lean VAT EVs**		
PTPRC_RAT	P04157	Receptor-type tyrosine-protein phosphatase C	0.002	2.82
PROF1_RAT	P62963	Profilin-1	0.047	2.67
PLST_RAT	Q63598	Plastin-3	0.050	2.46
TCPA_RAT	P28480	T-complex protein 1 subunit alpha	0.037	2.46
LETM1_RAT	Q5XIN6	Mitochondrial proton/calcium exchanger protein	0.030	2.30
RL8_RAT	P62919	60S ribosomal protein L8	0.055	2.27
NEDD4_RAT	Q62940	E3 ubiquitin-protein ligase NEDD4	0.049	2.24
DEST_RAT	Q7M0E3	Destrin	0.026	2.13
PP2AB_RAT	P62716	Serine/threonine-protein phosphatase 2A catalytic subunit beta isoform	0.018	2.09
FIS1_RAT	P84817	Isoform 2 of mitochondrial fission 1 protein	0.007	2.09
PGRC1_RAT	P70580	Membrane-associated progesterone receptor component 1	0.032	2.08
FUCO_RAT	P17164	Tissue alpha-L-fucosidase	0.004	2.07
SAHH_RAT	P10760	Adenosylhomocysteinase	0.053	2.04
ATPB_RAT	P10719	ATP synthase subunit beta, mitochondrial	0.053	2.03
PUR9_RAT	O35567	Bifunctional purine biosynthesis protein ATIC	0.043	1.98
		**Elevated in Obese VAT EVs**		
NHRF2_RAT	Q920G2	Na(+)/H(+) exchange regulatory cofactor NHE-RF2	0.034	9.64
K2C7_RAT	Q6IG12	Keratin type II cytoskeletal 7	0.035	4.41
S10A4_RAT	P05942	Protein S100-A4	0.001	3.70
PEBP1_RAT	P31044	Phosphatidylethanolamine-binding protein 1	0.027	3.49
VDAC2_RAT	P81155	Voltage-dependent anion-selective channel protein 2	0.030	3.00
S10A6_RAT	P05964	Protein S100-A6	0.003	2.93
ISOC1_RAT	Q6I7R3	Isochorismatase domain-containing protein 1	0.017	2.87
S100B_RAT	P04631	Protein S100-B	0.018	2.78
CLYBL_RAT	Q5I0K3	Citramalyl-CoA lyase. mitochondrial	0.019	2.67
DRS7B_RAT	Q5RJY4	Dehydrogenase/reductase SDR family member 7B	0.027	2.65
UCP1_RAT	P04633	Mitochondrial brown fat uncoupling protein 1	0.053	2.64
ALBU_RAT	P02770	Albumin	0.038	2.59
ODBB_RAT	P35738	2-oxoisovalerate dehydrogenase subunit beta, mitochondrial	0.029	2.39
ILEUA_RAT	Q4G075	Leukocyte elastase inhibitor A	0.016	2.33
CBPQ_RAT	Q6IRK9	Carboxypeptidase Q	0.049	2.29
S10AB_RAT	Q6B345	Protein S100-A11	0.030	2.15
VTDB_RAT	P04276	Vitamin D-binding protein	0.031	2.08
PURA_RAT	P86252	Transcriptional activator protein Pur-alpha (fragments)	0.021	2.07
K2C8_RAT	Q10758	Keratin type II cytoskeletal 8	0.005	2.05
HIBCH_RAT	Q5XIE6	3-hydroxyisobutyryl-CoA hydrolase mitochondrial	0.025	2.02
		**Elevated in Lean BAT EVs**		
RAB14_RAT	P61107	Ras-related protein Rab-14	0.018	4.19
QCR6_RAT	Q5M9I5	Cytochrome b-c1 complex subunit 6, mitochondrial	0.014	3.21
DX39A_RAT	Q5U216	ATP-dependent RNA helicase DDX39A	0.039	2.90
MMSA_RAT	Q02253	Methylmalonate-semialdehyde dehydrogenase (acylating), mitochondrial	0.019	2.69
ARP3_RAT	Q4V7C7	Actin-related protein 3	0.011	2.46
RS12_RAT	P63324	40S ribosomal protein S12	0.053	2.44
ACLY_RAT	P16638	ATP-citrate synthase	0.010	2.34
CPT2_RAT	P18886	Carnitine O-palmitoyltransferase 2, mitochondrial	0.008	2.31
ACADM_RAT	P08503	Medium-chain specific acyl-CoA dehydrogenase, mitochondrial	0.047	2.13
THIL_RAT	P17764	Acetyl-CoA acetyltransferase, mitochondrial	0.006	2.12
PYC_RAT	P52873	Pyruvate carboxylase, mitochondrial	0.003	2.10
IDH3A_RAT	Q99NA5	Isocitrate dehydrogenase [NAD] subunit Alpha, mitochondrial	0.027	2.07
HSP74_RAT	O88600	Heat shock 70 kDa protein 4	0.044	2.04
FETA_RAT	P02773	Alpha-fetoprotein	0.029	2.01
DECR_RAT	Q64591	2,4-dienoyl-CoA reductase [(3E)-enoyl-CoA-producing], mitochondrial	0.033	2.00
ODP2_RAT	P08461	Dihydrolipoyllysine-residue acetyltransferase component of pyruvate dehydrogenase complex, mitochondrial	0.010	1.99
		**Elevated in Obese BAT EVs**		
UCP1_RAT	P04633	Mitochondrial brown fat uncoupling protein 1	0.012	4.38
DPEP1_RAT	P31430	Dipeptidase 1	0.003	3.71
PCKGC_RAT	P07379	Phosphoenolpyruvate carboxykinase, cytosolic [GTP]	0.055	3.42
FAHD1_RAT	Q6AYQ8	Acylpyruvase FAHD1, mitochondrial	0.001	3.26
FUCO_RAT	P17164	Tissue alpha-L-fucosidase	0.009	3.18
REEP5_RAT	B2RZ37	Receptor expression-enhancing protein 5	0.002	3.07
CD9_RAT	P40241	CD9 antigen	0.014	3.03
UCRI_RAT	P20788	Cytochrome b-c1 complex subunit Rieske, mitochondrial	0.027	2.89
TERA_RAT	P46462	Transitional endoplasmic reticulum ATPase	0.002	2.75
2|RTN3_RA	Q6RJR6	Isoform 2 of reticulon-3	0.004	2.72
SEC13_RAT	Q5XFW8	Protein SEC13 homolog	0.040	2.72
AOC3_RAT	O08590	Membrane primary amine oxidase	0.035	2.69
TARG1_RAT	Q2MHH0	Trafficking regulator of GLUT4 1	0.021	2.68
LEG5_RAT	P47967	Galectin-5	0.007	2.59
GLPK_RAT	Q63060	Glycerol kinase	0.002	2.59
CLH1_RAT	P11442	Clathrin heavy chain 1	0.004	2.56
ECI1_RAT	P23965	Enoyl-CoA delta isomerase 1, mitochondrial	0.017	2.54
PARVA_RAT	Q9HB97	Alpha-parvin	0.005	2.43
AT1B3_RAT	Q63377	Sodium/potassium-transporting ATPase subunit beta-3	0.006	2.41
PP2AB_RAT	P62716	Serine/threonine-protein phosphatase 2A catalytic subunit beta isoform	0.006	2.40
SUOX_RAT	Q07116	Sulfite oxidase, mitochondrial	0.016	2.39
CD36_RAT	Q07969	Platelet glycoprotein 4	0.003	2.38
FAAA_RAT	P25093	Fumarylacetoacetase	0.001	2.35
MVP_RAT	Q62667	Major vault protein	0.004	2.31
HYEP_RAT	P07687	Epoxide hydrolase 1	0.011	2.30
NIT2_RAT	Q497B0	Omega-amidase NIT2	0.012	2.30
GRIFN_RAT	O88644	Grifin	0.006	2.29
SAP_RAT	P10960	Prosaposin	0.009	2.29
5NTD_RAT	P21588	5′-Nucleotidase	0.019	2.27
PGAM1_RAT	P25113	Phosphoglycerate mutase 1	0.001	2.24
DPP2_RAT	Q9EPB1	Dipeptidyl peptidase 2	0.000	2.24
MDHM_RAT	P04636	Malate dehydrogenase, mitochondrial	0.047	2.22
PLPP3_RAT	P97544	Phospholipid phosphatase 3	0.030	2.20
ACOC_RAT	Q63270	Cytoplasmic aconitate hydratase	0.004	2.19
SBP1_RAT	Q8VIF7	Methanethiol oxidase	0.041	2.19
PLBL1_RAT	Q5U2V4	Phospholipase B-like 1	0.005	2.16
LDHB_RAT	P42123	L-lactate dehydrogenase B chain	0.000	2.15
AP1B1_RAT	P52303	AP-1 complex subunit beta-1	0.031	1.97
GPDA_RAT	O35077	Glycerol-3-phosphate dehydrogenase [NAD(+)], cytoplasmic	0.008	1.97
RAC1_RAT	Q6RUV5	Ras-related C3 botulinum toxin substrate 1	0.016	1.97
RL10A_RAT	P62907	60S ribosomal protein L10a	0.003	1.97

^1^ Selected proteins with a fold change of 2 or bigger; the complete list with a fold change higher than 1.5 is displayed in Appendix A.

## Data Availability

Not applicable.

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
