# Peer review of "Brown Adipose Tissue Sheds Extracellular Vesicles That Carry Potential Biomarkers of Metabolic and Thermogenesis Activity Which Are Affected by High Fat Diet Intervention"

_ijms, 2022, doi:10.3390/ijms231810826_

Round 1

Reviewer 1 Report

In this paper, the authors have performed an extensive analysis of those vesicles secreted by adipose tissue at different anatomical locations elucidating the EVs cargo changes after high fat diet intervention. Besides, the authors isolated those vesicles shed by brown adipose tissue in the same animals describing, the reference proteome map of batosomes and additionally performing a quantitative SWATH proteome analysis in comparison with white adipose tissue-shed vesicles (subcutaneous and visceral depots) from the same animals under lean and obese condition. Functional analysis and proteins of interest validation reveals batosomes protein cargo dynamics during the development of obesity. This study is novel and of great interest. However, there are several issues that still need improvement.

1. All abbreviations in the manuscript must be written in full when first mentioned in the text and subsequently, the use of abbreviations is okay.

2. p in statistical analysis should be italicized.

3. There should be a summary at the end of each section of results to clearly summarize the main findings of the section.

4. Line 135: g/cm2 ? or g/cm2 ?

5. Line 156: Please delete the extra , .

6. Lines 158-162: Are the expressions of  CD81+ CD9+ CD63+ correct ?

7. Please abbreviate high fat diet in the first reference, not in line 177.

8. There are some results in the manuscript that are not statistically significant (such as, characterization of isolated vesicles, the size of EVs, ), so are the statements about the conclusions informative? This is confusing to us.

9. We recommend that the control SAT in figure 8 be rendered in a darker color. The same problem occurs in the SAT of figure 9.

10. In this study, different numbers of animals used in different experimental sections, does this cause differences? Is it possible to keep the number of animals consistent? Please explain the reason.

11. Line 494: “ kg/m2” ? or “ kg/m2 ?

12. Line 784: CO2 ? or CO2 ?

13. Lines 746, 759, 760, 763: minutes ? or min ?

14. Please add the specific operation method for understanding (“Isolation of EVs by differential ultracentrifugation”“Immunoblotting”Nanoparticle Tracking Analysis).

15. Checking the full manuscript carefully is necessary. This paper had a few grammatical errors and editing issues. Therefore, the writing must be improved before resubmission.

Author Response

Reviewer 1

  1. All abbreviations in the manuscript must be written in full when first mentioned in the text and subsequently, the use of abbreviations is okay.

We have revised the whole manuscript and written the abbreviations in full when first mentioned in the text.

  1. “p” in statistical analysis should be italicized.

This has been corrected.

  1. There should be a summary at the end of each section of results to clearly summarize the main findings of the section.

Although we tried to summarize the main results on each paragraph title, we agree with the reviewer and added summaries in each section.

  1. Line 135: “g/cm2” ? or “g/cm2” ?

This has been corrected.

  1. Line 156: Please delete the extra “,” 

This has been corrected.

  1. Lines 158-162: Are the expressions of  “CD81+” 、“CD9+” 、“CD63+” correct ?

We agree with the reviewer that these expressions may be confusing; we have changed all + for “positive”

  1. Please abbreviate “high fat diet” in the first reference, not in line 177.

We have abbreviated HFD in the first reference of each section (introduction, results, conclusions…) to make it easier to the reader.

  1. There are some results in the manuscript that are not statistically significant (such as, characterization of isolated vesicles, the size of EVs, ), so are the statements about the conclusions informative? This is confusing to us.

We agree with the reviewer about this issue; however, please note that the limited amount of vesicles isolated from each whole adipose tissue is characteristically limiting their analysis. We decide to allocate the greater number of animals and its EVs for the proteomics and immunoblot validation analyses as this is the main purpose of our work. Our previous and extensive experience analyzing EVs from human and murine samples (Camino et al., 2020, 2021) has showed us that there are no significant changes when characterizing EVs by NTA, TEM or Exoview (Nanoview) platforms; that is why we decide to keep most of the samples for protein characterization and validation. Nevertheless, Exoview analysis permitted to perform statistical analysis of size and amount of vesicles confirming NTA results.

  1. We recommend that the “control SAT” in figure 8 be rendered in a darker color. The same problem occurs in the “SAT” of figure 9.

It is true that the chosen color is too light; we have changed this accordingly as requested.

  1. In this study, different numbers of animals used in different experimental sections, does this cause differences? Is it possible to keep the number of animals consistent? Please explain the reason.

We agree that this may be confusing; however, when working with EVs, the amount of vesicles isolated from adipose tissue is very limited and normally it is impossible to use the same sample for different experiments; that is why we had to share isolated EVs from each tissue from different animals among different experiments. Moreover, technical problems such as tissue culture contamination or poor vesicle isolation during ultracentrifugation is common and a limitation on this type of studies.

  1. Line 494: “ kg/m2” ? or “ kg/m2” ?

Please see that this has been corrected.

  1. Line 784: “CO2” ? or “CO2”?

Please check that this has been mended.

  1. Lines 746, 759, 760, 763: “minutes” ? or “min” ?

This has been corrected.

  1. Please add the specific operation method for understanding (“Isolation of EVs by differential ultracentrifugation” “Immunoblotting”, “Nanoparticle Tracking Analysis”).

We kindly thank the reviewer for this comment that made us realize that the operation method was not well explained in Figure 1 and in the first paragraph of the results. We have clarified this issue by clearly indicating the followed procedures in order: isolation of EVs by ultracentrifugation followed by EVs analysis by NTA (for size), mass spectrometry (for protein identification), and immunoblot (for protein validation). In the case of Exoview, EVs need no previous isolation by ultracentrifugation; therefore, samples were directly analyzed on the functionalized chips. Exoview (Nanoview platform) allows vesicle multi-analysis permitting to count the number of vesicles, obtain vesicles size by interferometry, and detect and quantify tetraspanins through fluorescent antibodies. Please see that we have modified Figure 1 and the text accordingly.

  1. Checking the full manuscript carefully is necessary. This paper had a few grammatical errors and editing issues. Therefore, the writing must be improved before resubmission.

Please see that we have revised the whole manuscript for grammatical errors by using an english editor specialist. 

Reviewer 2 Report

The authors have presented detailed comparative analysis of extracellular vesicles (EVs) from lean and obese adipose adipose tissues (BAT) primarily focusing on the brown adipose tissue. As an extension of their ongoing work, the authors adopted very similar comprehensive approach to identify and characterize visceral (VAT), subcutaneous (SAT) and BAT EVs from 5 week-old lean and obese male rats after 9 week (short-term) feeding of high fat diet. 

While this manuscript present some very interesting and novel findings, they need to consider several points and address the appropriately.   

Major points:

1. The manuscript should be shortened and compact to convey the key findings of the study. More specifically, some figures could either be combined or eliminated without altering the important messages.

2. Considering a profound gender difference in adipose tissue response to various stress-inducing stimuli including diet-induced obesity (DIO), the relevance of these findings in female mice only is not clear.

3a. The choice of short term high fat diet (9 weeks) is not clear. The possible time-sensitive effects of this study on adipose EVs as compared to 20 week high fat diet used in Alcala et. al. studies (showing increased levels of UCP1 protein and not gene) expression could be profound and are not comparable.

3b. Functional characterization of BAT and VAT EVs (including but not limited to cellular respiration and mitochondrial content) from lean and obese rats after short term DIO to correlate the relevance of observed increase in UCP1 protein in obese group is needed.  

3c. Expression levels of key BAT/beige-selective markers in all EVs obtained from adipose tissue groups should be helpful to further explore/confirm the observed increase in UCP1 levels in obese group. 

4. Did the authors analyzed the relative abundance of few highly relevant proteins including adiponectin (highly expressed in all adipose tissue depots and a major responder of obesogenic conditions to influence glucose and lipid metabolism in their analyses? 

Minor points:

1. Figure 2D: Not clear which bars represent CD9 and CD81.

2. Reference 23 -incomplete details

Author Response

Reviewer 2

  1. The manuscript should be shortened and compact to convey the key findings of the study. More specifically, some figures could either be combined or eliminated without altering the important messages.

We agree with the reviewer and we have made significant changes to shorten and compact the manuscript: please check that we have combined figures 4 and 5 in one: now Figure 4 in the new version of the manuscript. Therefore, we have grouped those quantitative DIA-SWATH/MS analysis of white adipose tissues in just one figure. On the other hand, we decided to move figure 8 representing the area values obtained by DIA-SWATH/MS analysis to supplementary figures (now Figure S3), as it is true that this information may be redundant in the article.

  1. Considering a profound gender difference in adipose tissue response to various stress-inducing stimuli including diet-induced obesity (DIO), the relevance of these findings in female mice only is not clear.

We totally agree with the reviewer, and indeed it is our intention to expand our research in this sense as we and our institution are committed to gender equality in experimental research. Please note that our previous research on human obese adipose tissue EVs was performed in female patients as they are the most prone to bariatric surgery; that is the reason why we have compared the results in this work with our previous human and murine cell lines EVs characterization. The fact that we have many protein/biomarkers in common to our previous research shows that this work may be illustrative of EVs content, however, it is true that gender must be taken into account and female animals will be analyzed in the near future.

3a. The choice of short term high fat diet (9 weeks) is not clear. The possible time-sensitive effects of this study on adipose EVs as compared to 20 week high fat diet used in Alcala et. al. studies (showing increased levels of UCP1 protein and not gene) expression could be profound and are not comparable.

We agree; we just intended to show that our results are coherent with previous reports showing stable or elevated levels of UCP1 after HFD intervention. Please note that we have indicated this in the discussion section of the new version of the manuscript.

3b. Functional characterization of BAT and VAT EVs (including but not limited to cellular respiration and mitochondrial content) from lean and obese rats after short term DIO to correlate the relevance of observed increase in UCP1 protein in obese group is needed.  

Please note that functional characterization for all identified proteins from all tissue depots and conditions were not limited in any case. We performed FunRich and Reactome analysis with all identified proteins; this type of software allows the identification of overrepresented classes of genes or proteins by comparing against annotated databases. The fact that certain classes of proteins were predominant, such as cellular respiration or mitochondrial content, is given by the algorithm analysis, we did not conduct or limited the analysis at all. Nevertheless, please note that the whole functional analysis results for each tissue depot and condition is shown in Supplementary tables and in Appendix A and B of the manuscript.

3c. Expression levels of key BAT/beige-selective markers in all EVs obtained from adipose tissue groups should be helpful to further explore/confirm the observed increase in UCP1 levels in obese group. 

We agree; please note that this information is detailed now in Table S11 describing the results of SWATH analysis comparing obese BAT identified proteins to all the rest depots; precisely, we have highlighted UCP1 fold change in this table.

  1. Did the authors analyzed the relative abundance of few highly relevant proteins including adiponectin (highly expressed in all adipose tissue depots and a major responder of obesogenic conditions to influence glucose and lipid metabolism in their analyses? 

Indeed, many relevant proteins influenced by obesogenic conditions are shown in our work in both qualitative and quantitative analysis. They can be found in the Supplementary tables; some examples are: caveolins (1/2), FAS, FABP4, MIF, lipoprotein lipase, perilipin, which are all elevated in obese white adipose tissue (please see Suppl. Tables 7-9).

Minor points:

  1. Figure 2D: Not clear which bars represent CD9 and CD81.

Please note that bars show the sum of captured particles by both tetraspanins to get the total amount of captured particles.

  1. Reference 23 -incomplete details

This has been corrected.

Round 2

Reviewer 1 Report

Accept in present form

Reviewer 2 Report

The authors have responded to the comments and made appropriate changes in the revised manuscript. I have no additional comments.